# Photovoltaic Energy Harvesting System Adapted for Different Environmental Operation Conditions: Analysis, Modeling, Simulation and Selection of Devices

**DOI:** 10.3390/s19071578

**Published:** 2019-04-01

**Authors:** Borja Pozo, José Ignacio Garate, José Ángel Araujo, Susana Ferreiro

**Affiliations:** 1Electronics and Communications Unit and Intelligent Information System Unit, IK4-Tekniker, Calle Iñaki Goenaga 5, 20600 Eibar, Spain; susana.ferreiro@tekniker.es; 2Department of Electronics Technology, University of the Basque Country (UPV/EHU), 48080 Bilbao, Spain; joseignacio.garate@ehu.eus (J.I.G.); joseangel.araujo@ehu.es (J.A.A.)

**Keywords:** energy harvesting, photovoltaic harvester, DC/DC converter, environmental conditions, simulation

## Abstract

The present research work proposes a photovoltaic energy harvester and an appropriate direct current (DC)/DC converter for a harvesting system after the study of the devices and taking the operation conditions. Parameters such as power, efficiency and voltage are taken into account under different environment conditions of illumination and temperature in order to obtain the best possible response. For this reason, suitable metal-oxide semiconductor field-effect transistor (MOSFET), diode, coil, frequency, duty-cycle and load are selected and analyzed for a DC/DC converter with boost architecture.

## 1. Introduction

Nowadays, low-cost, smart electronic systems and wireless sensor network technologies are experiencing fast growth. They are currently being deployed worldwide into both home [1] and industry environments [2,3]. On the one hand, they will improve the quality of people’s lives: environment, health and well-being, security, comfort, education and entertainment [4]. On the other hand, future smart industry and factories will become more efficient, intelligent, and connected [5]. This is due to the fact that smart sensors, with embedded intelligence for controlling, communication and interoperability can be integrated within enterprise business systems, thus constituting a change of system architecture in automation and control process [3].

Besides, there is an increasing interest in green electronics [6] and among other characteristics, for an electronic system to be green, it must have a contained price and be energy efficient [7]. Thus, electronic devices or systems with wireless capabilities are increasingly popular because they do not need to be connected to the mains power grid [8]. In this context, energy harvesters become a proper alternative for gathering energy from the environment and provide answers to some of the aforementioned technical challenges [9,10], because energy harvesters are, essentially, transducers devised to extract, not only a sample of the physical phenomena the aim for, but the maximum feasible amount of energy [11].

A generic energy harvesting system has three main elements [12]: a harvester, low power management, and a low power storage system. Once the harvester is selected, its characteristics determine whether direct current (DC)/DC or alternating current (AC)/DC conversion is required [13]. Afterwards, specifications of its architecture and devices should be analyzed. Many harvesters collect energy in AC form, while a few, such as photovoltaic and thermoelectric provide DC signals. Consequently, the converter of a harvester must be adapted to the energy and waveforms that its technology provides. Therefore, it requires specific research on each of the elements that constitutes a complete energy harvester [14,15].

This research work is devoted to the characterization, modelling, design and parametrization of DC harvesters, specifically, photovoltaic harvesters and their required power converter. Firstly, the model for the energy harvester, a photovoltaic cell, is defined and analyzed through and mixed structural and electrical model and experimental results. Based on the previous results, the DC/DC converter architecture is selected. The research ends with a statistical modelling and analysis of the performance of the state-of-the-art passive and active devices within the DC/DC converter architecture taking in account parameters as power, efficiency, voltage and current waveforms. The results obtained are used to identify the most suitable arrangement of discrete components for maximum efficiency, performance and given by the DC/DC converter architecture.

## 2. Energy Harvester: Photovoltaic Cell

The analyzed harvester is a photovoltaic cell, which provides a DC signal [16]. Before the energy can be used, it has to go through several intermediate steps. At an early stage, light power is harvested and converted to electrical power with an efficiency η_photovoltaic_cell_ [17]. Then, the level of the harvested DC signal is adequated to an appropriate level for tis storage. The latter is done through a DC/DC converter, with an efficiency of η_converter_ [18]. So, the energy is ready for the storage system: battery [19]. or supercapacitor [20].

Figure 1 shows the DC system block diagram and energy conversions with respective efficiencies:

Equation (1) provides the power balance as the product of the systems conversion efficiencies and the input light power: (1)$$P_{3}=\eta_{\mathrm{tot}}\cdot P_{1}=\left(\eta_{\mathrm{photovoltaic}\_\mathrm{cell}}\cdot \eta_{\mathrm{converter}}\right)\cdot P_{1}=\left(\eta_{1}\cdot \eta_{2}\right)\cdot P_{1}$$

### 2.1. Photovoltaic Harvester Model

The definition of an equivalent model of a photovoltaic cell has been based on the work of [21,22]. The chosen photovoltaic harvester is based on a five-parameter Equation (2):(2)$$I=I_{\mathrm{ph}}-I_{o}\left(e^{\frac{q\left(V+{\mathrm{IR}}_{s}\right)}{{\mathrm{nkT}}_{c}}}-1\right)-\frac{V+{\mathrm{IR}}_{s}}{R_{\mathrm{sh}}}$$
where, k is the Boltzmann constant, q is the electron charge, I_ph_ is the current generated by the light, Io is the dark saturation current due to recombination, R_s_ is a series resistance, n is the ideal factor, T_c_ is the cell temperature, and R_sh_ is a shunt resistance. This model is a function of the solar radiance G and the air temperature T_a_. Equation (2) can be used to measure the value of the parameters I_ph_, I_o_, n, R_s_, R_sh_ under real-time ambient conditions.

Figure 2 represents the equivalent electric circuit of a single photovoltaic cell. The circuit consists of a main circuit and two sub-circuits to modify R_sh_ and R_s_ values when the ambient condition changes, these represent the calculation of the real-time resistors. 

The main circuit includes a current source, a diode, two opposite voltage sources and cell dissipate phenomena equivalent resistors which represents the behavior of the cell in open circuit state. The current generated by the light source is modelled through two current sources and a resistor in shunt, which is irradiance dependent. Temperature effects are included as a two-voltage source in series with opposite polarities. The unconnected point at the main circuit represents the output of the cell, the remaining parts are for the configuration of parameters and variables. In order to complete the model, an electric load is connected in parallel with the voltage output of the two branches.

The photovoltaic model is completed with the manufacturer data-sheet and the calculated parameters. The solar harvester devised for this research uses the solar module SLMD481H08 manufactured by IXOLARTM [23].

The selected monocrystalline module has 88.8 mm of width and 54.7 mm of height. Each individual cell has 20 × 12 mm^2^ = 2 × 1.2 cm^2^ = 2.4 cm^2^. The whole module has 16 cells distributed in two branches connected in parallel, each one made of 8 cells. The cell has a nominal efficiency of 22%. It also has a broad photonic response over a wide range of wavelengths. Thus, it is suitable for both applications, indoor and outdoor. The architecture and dimensions of the module used is shown in Figure 3.

Electrical characteristics of the selected solar module provided by the manufacturer are presented in Table 1.

The whole photovoltaic module model is made of single cells (Figure 2) arranged as a commercial selected photovoltaic module, Figure 3: 16 main cells in 2 rows/branches of 8 cells each. Simulations provide the maximum out power for the selected ambient conditions [25]. Table 2 and Figure 4 show the achieved results.

These results show that the cells produce more power at higher temperatures for the same irradiance level. Besides, irradiance has more effect on power production than temperature, e.g., 0.25 sun and 10 °C against 0.1 sun 25 °C. The cell power generation range from 22.333 mW with 0.1 sun at 5 °C to 306.812 mW with 1 sun at 60 °C.

### 2.2. Model Simulation and Test Results Comparison

In order to verify and validate the accuracy of the photovoltaic electro-physical model, solar tests are carried out. This test is done with a sun luminaire, a controlled source of light. This solar light had the restriction to produce a maximum of 0.5 sun. In addition, a black box is used to avoid any interference of irradiance on the test and maintain the light seal. Figure 5a shows the test set-up diagram, and Figure 5b shows the real implementation of the set-up.

The test procedure has been as follows: The solar module is positioned inside the black box and under the luminaire. The electronic load is connected to the solar cell, for a sweep measurement. A data logger is connected to the solar cell and electronic load in order to measure and register the voltage and current produced by the solar cell. The temperature sensor, PT-100, is located in the box near the solar cell and connected to the temperature measurement system, configured to save data periodically. Finally, the test stars and data is registered for future processing.

Some tests under different irradiance and temperature conditions are developed to verify the model and two of them are shown below under the following conditions:
Solar irradiance 0.25 sun and 46 °C temperature.Solar irradiance 0.5 sun and 65 °C temperature.

Then, these tests are compared with electro-physical model simulations. In Figure 6 and Figure 7, blue traces represent the simulation results and the orange traces the values measured during the test.

With 0.25 sun and 46 °C, short circuit current is 27 mA, Figure 6a; MPPT (maximum power point track) is 75 mW and is produced around 3.4 V, Figure 6b.

With 0.5 sun and 65 °C, short circuit current is 78 mA, Figure 7a; MPPT is 180 mW and is produced around 2.9 V, Figure 7b. 

In order to verify the model, the deviation between the model values and the data measured is determined, and the results are shown in Table 3.

In those tests the worst deviation, which is 3.148% in current, occurs when ambient conditions are G = 0.25 sun and T = 46 °C, with 3.148% in current and 3.009% in potency. The lower deviation is produced in G = 0.5 sun and T = 65 °C test, with 2.316% in current and 2.395 in potency. The obtained results confirm that the model is appropriate and can be used in the research.

## 3. Direct Current (DC)/DC Converter

Usually, the DC/DC converter is placed between the harvester and the storage device as the power manager. The system block diagram with the harvester (photovoltaic cell), power management (DC/DC converter) and the storage device (supercapacitor) is shown in Figure 8.

The main objective of this subsection is to research, simulate and verify the DC/DC converter for the photovoltaic cell, considering all operating voltage and frequency ranges, and assuming an efficiency greater than 80%.

The DC/DC converter requirements are:Impedance matching between source and load for maximum energy transference.To step up the voltage received from harvesting.To maintain operation functionality and efficiency with sun irradiance as well as with artificial light irradiance.MPPT control algorithm.

### 3.1. Boost Architecture and Control Definition

#### 3.1.1. Boost Architecture

The architecture employed in the DC/DC converter is the boost topology [26,27] showed in Figure 9. The main challenge of the design of this converter is that it must manage low power levels and voltage [28], which limits the performance and availability of the semiconductors.

Boost control techniques are based on the change of the duty cycle value. This element controls the steady-state output concerning the input voltage [29]. The DC/DC converter construction starts with the selection of the coil value, MOSFET, and diode appropriate devices. This selection has been made bearing in mind two extreme temperature and illumination operation conditions:
Minimum: G = 0.1 sun and Temperature = 5 °C.Maximum: G = 1 sun and Temperature = 60 °C.

The selection of the coil value and MOSFET and diode devices was performed with Linear Technology Simulation Program with Integrated Circuits Emphasis (LTSPICE^®^) [30], however other simulation packages could be used.

#### 3.1.2. Control Definition

Nowadays, most converters have two modes of operation or control schemes, a continuous mode with pulse frequency modulation (PFM) or pulse wide modulation (PWM) control for maximum output power; i.e., maximum load and discontinuous control mode (DCM) with variable frequency for low loads or unloaded condition, which can be implemented through current or voltage control loops. The DCM improves the quiescent current of the converter, and, consequently, its efficiency [29]. Besides, most DC-DC converters for photovoltaic (PV) modules employ different types of MPPT control algorithms in order to drain, at any time, the maximum power that the PV module provides. The basic form of MPPT tracking algorithm employs the measure of input power and adjust the DC-DC control signal accordingly to get the maximum power the PV could deliver regardless the load demands, the eventual excess or lack of energy should be absorbed of provide by a backup storage device: battery or supercapacitor. In the literature could be found many types of MPPT algorithms [31]. Thus, the control of the DC-DC converter must include an MPPT control algorithm.

The switching frequency of the converter constitutes a key parameter to improve maximum efficiency, and it is related with the power input condition. Therefore, the goal is to minimize the switching losses in the converter, it is done reducing the frequency and choosing active devices that provide less switching losses. On the other hand, there is the issue of component size, the higher the frequency, the smaller the inductance and capacitor values and sizes will be, and consequently, the lower the losses in the inductance serial resistance and the voltage drop in Equivalent Series Resistance (ESR) of the capacitors. The aforementioned is achieved at the expense of increasing the switching losses.

In the present study, the DC-DC converter is of low power, therefore the control frequencies may range from 10 kHz to tenths of MHz, more concretely to hundreds of kHz are shown in Section 3.4. 

### 3.2. MOSFET Selection

At minimum temperature conditions and irradiance, the switching frequency is set to 50 kHz, the duty-cycle to 0.15, the coil inductance to 270 µH, the R_load_ to 400 Ω and the selected commercial diode model is the MBR0520L. Next, several commercial MOSFETs are selected to test on the basis of their appropriate characteristics for the present case in use: Table 4. At maximum ambient conditions, the switching frequency is set to 20 kHz, the duty-cycle to 0.16, the R_load_ to 50 Ω and the coil value and diode are maintained. Table 4, Table 5 and Figure 10 summarize the results with detailed conditions.

The results obtained show that the efficiency for both conditions is higher than 90% with all MOSFETs. Moreover, with low light irradiance and temperature they reach up to 91% and in high conditions up to 93%. Si1555DL_N and Si3900DV present the best efficiency regarding the rest of the devices. However, Si1555DL_N and Si1553CDLN have the least consumption at both work operation modes. Because of these reasons, Si1555DL_N MOSFET is selected to be the MOSFET at the DC/DC boost converter. However, Si3900DV is also a feasible alternative to be used.

### 3.3. Diode Selection

The diode is selected employing the same procedure as with the MOSFET, the same minimum circuit values are adopted. So, several commercial diodes are selected on the basis of their characteristics for the present case in use: Table 6. In Table 6, Table 7 and Figure 11 the results achieved with the conditions detailed are shown.

Efficiencies in this case at both conditions are higher than 90% with the exception of the BAT15-03W diode. Moreover, in low light and temperature ambient they reach up to 92% and in high conditions up to 95%. STPS20L15G and MBR0520L present the best efficiency regarding the rest of the diodes. However, with STPS20L15G converter, efficiency is 0.4% higher at the worst ambient conditions and 3% better in the best ambient conditions. Due to these reasons, the STPS20L15G diode is selected for the DC/DC boost converter. However, after having issues with the PSPICE model of the STPS20L15G diode, MBR0520L is used for simulations because it has a proper response too. Nevertheless, it is recommended to keep track continuously of the new releases of the diode manufacturer in order to consider STPS20L15G.

### 3.4. Coil and Switching Frequency Selection

With minimum ambient conditions, the duty-cycle is set to 0.2, the R_load_ to 2 kΩ and the selected diode in the previous section. In the Table 8, Table 9 and Figure 12 the results achieved with the conditions detailed are shown.

If input/output power is taken into account, the simulation results show that the most suitable value for the coil is 200 µH. With 0.1 sun and 5 °C temperature condition the best coil value is 100 µH and with 1 sun and 60 °C temperature the best coil is 300 µH. However, the coil values for maximum and minimum extreme conditions are not adequate for nominal conditions; consequently, 200 µH is an intermediate value which gives stability.

The performed test shows that 100 kHz provides the best performance at low input power levels, and 20 kHz provides the best performance for high input power levels. Thus, in order to obtain get the maximum efficiency from the converter, its switching frequency should be dynamically changed between the previous values depending on the ambient conditions and the power input ranges or a trade of the previous frequencies [45].

### 3.5. Simulations and Results of the Converter. Duty-Cycle and Resistive Load

After making the selection of the components and establishing operation modes of control, several simulations have been carried out to analyze, adjust and obtain the maximum efficiency of the converter at different ambient conditions and input power levels.

The MPTT control algorithm of the DC-DC converter guaranties that the input power is delivered to the load with maximum efficiency, but the maximum power transfer occurs when the source and the load are matched, which is not always feasible. In solar farms this issue does not exists for the electrical grid absorbs the fluctuations. Therefore, in order to further improve the energy efficiency, it should be required a technique to dynamically change the load. This could be achieved by means of an energy storage device, battery or supercapacitor, and controlling the energy it stores or deliver. In doing so, it is required to know the load consumption patterns and the optimum loads of the PV and DC-DC converter, [46] analyses DC-DC power supply systems for pulsed loads from maximum efficiency perspective.

In order to identify the optimum values of load and duty-cycle, and dimension, accordingly, the energy storage device, a load and duty-cycle sweep is performed in five ambient condition ranges. Figure 13 shows the sweep results obtained.

In the first case (Figure 13a), the highest output power is 21.247 mW and is obtained with 0.2 duty-cycle and 500 Ω resistive load. In the second case (Figure 13b), the highest output power is 61.476 mW and is obtained with 0.3 duty-cycle and 400 Ω resistive load. In the third case (Figure 13c), the highest output power is 122.292 mW and is obtained with 0.3 duty-cycle and 150 Ω resistive load. In the fourth case (Figure 13d), the highest output power is 195.57 mW and is obtained with 0.3 duty-cycle and 100 Ω resistive load. In the fifth case (Figure 13e), the highest output power is 283.18 mW and is obtained with 0.2 duty-cycle and 50 Ω resistive load. 

Table 10 and Figure 14 summarize maximum output values obtained in previous simulations for different irradiance and temperature levels: from 0.1 sun and 5 °C to 1 sun and 60 °C.

Table 10 and Figure 14a show the evolution of the input (V_in_) and output (V_out_) voltage levels at the different environment conditions considered. The DC/DC converter provides an output voltage between 3.7 V and 4.9 V, this range is into the most common range of power supply voltage in consumer electronics. The results in Figure 14b show that the power increases linearly with irradiance. Moreover, whenever the input irradiance power increases, so does the difference between input and output power because the system efficiency diminishes. Besides, Figure 14c shows that the efficiency is always much greater than the 80% considered in Section 2 for the DC/DC converter architecture, which correspond to the expected efficiency of commercial boost converters [47,48]. The highest efficiency is 94.03%, and occurs at 0.5 sun and 15 °C; and the lowest efficiency (92.30%), is obtained at 0.75 sun and 40 °C.

## 4. Conclusions

Energy harvesters are being increasingly used for gathering energy from the environment and feeding electronic circuits without the help of a mains power grid connection or battery back-up support. An energy harvester system needs management control, and in this work a DC/DC converter is proposed for a photovoltaic harvester after the analysis of different possible devices, looking at different parameters such as efficiency, power and voltage.

A boost architecture is chosen for the DC/DC converter, and between considered devices the chosen ones are: the MOSFET Si1555DL_N (Si3900DV is adequate too), the diode STPS20L15G (MBR0520L is adequate too), the suitable value for the coil is 200 µH, the switching frequency must be dynamic between 150 kHz and 20 kHz depending on the input condition. Finally, the appropriate duty-cycle and resistive load for each input case are analyzed, the best results are obtained with a duty-cycle between 0.2 and 0.3, and the load between 50 Ω and 500 Ω.

The results show that proposed harvesting system works properly for the considered ambient conditions.

## Figures and Tables

**Figure 1 sensors-19-01578-f001:**
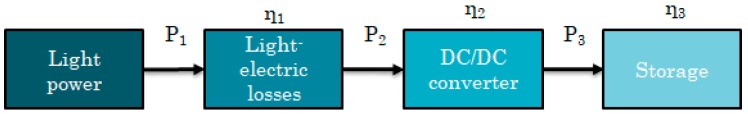
Block diagram of complete DC system.

**Figure 2 sensors-19-01578-f002:**
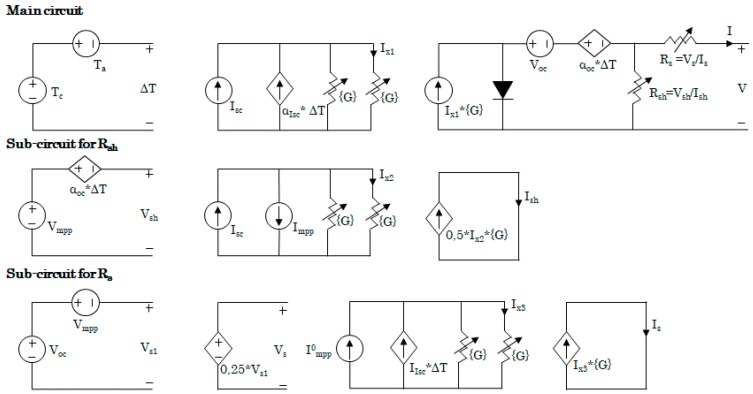
Single cell equivalent electrical model.

**Figure 3 sensors-19-01578-f003:**
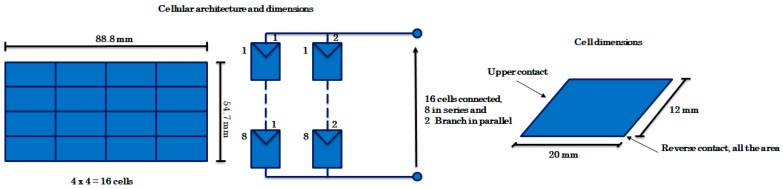
Cellular architecture and dimensions used.

**Figure 4 sensors-19-01578-f004:**
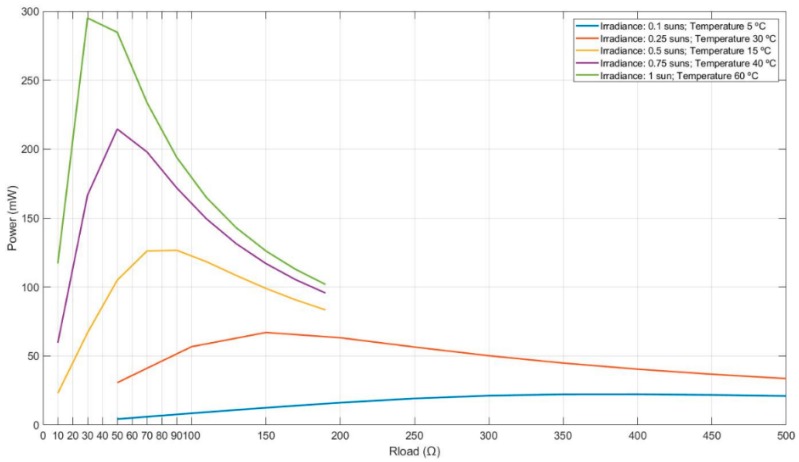
Simulated power curves with load sweep for 0.1, 0.25, 0.5, 0.75 and 1 sun irradiance at different temperatures.

**Figure 5 sensors-19-01578-f005:**
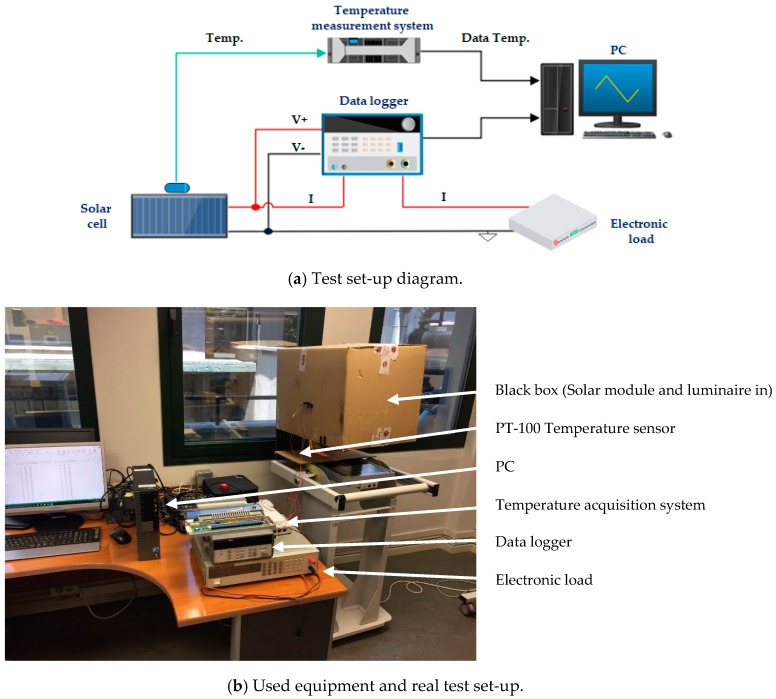
Photovoltaic cell test set-up.

**Figure 6 sensors-19-01578-f006:**
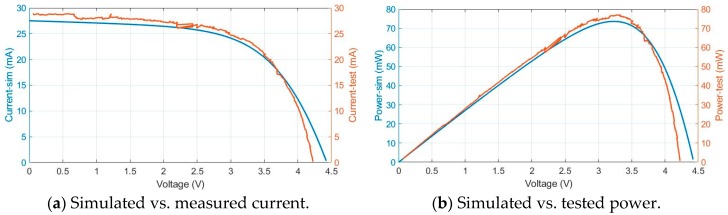
Test vs. simulation photovoltaic module results with 0.25 sun irradiance and 46 °C temperature.

**Figure 7 sensors-19-01578-f007:**
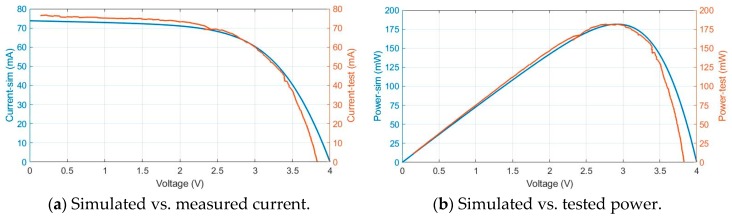
Test vs. simulation photovoltaic module results with 0.5 sun irradiance and 65 °C temperature.

**Figure 8 sensors-19-01578-f008:**
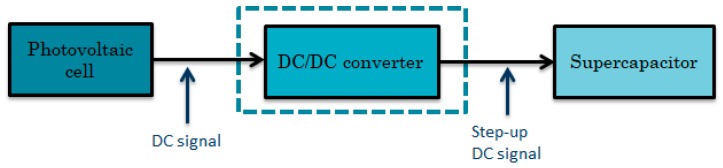
DC complete system block diagram.

**Figure 9 sensors-19-01578-f009:**
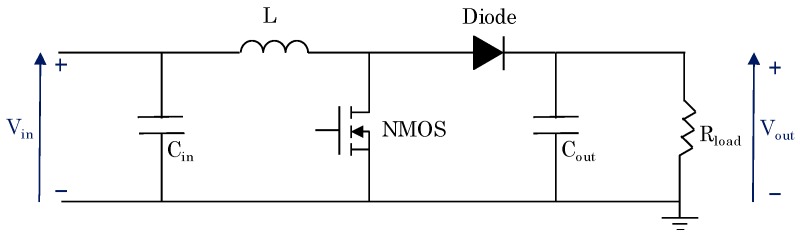
DC/DC boost architecture for the photovoltaic module converter.

**Figure 10 sensors-19-01578-f010:**
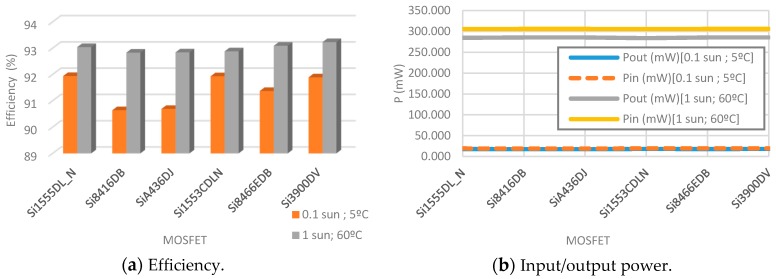
Comparisons between MOSFETs at 0.1 sun and 5 °C; and 1 sun and 60 °C.

**Figure 11 sensors-19-01578-f011:**
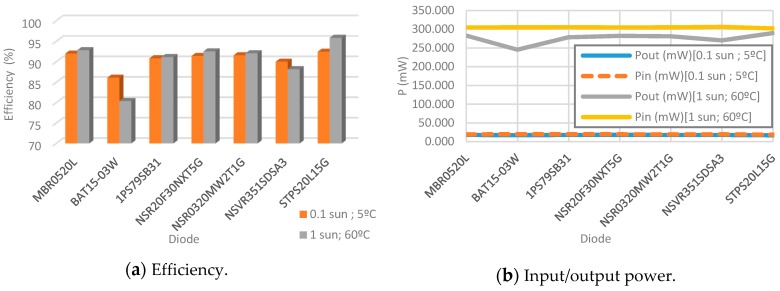
Comparisons between diodes at 0.1 sun and 5 °C; and 1 sun and 60 °C.

**Figure 12 sensors-19-01578-f012:**
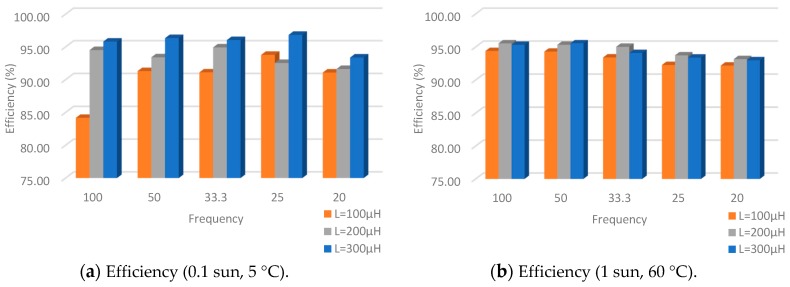
Comparisons between different coils and frequencies at 0.1 sun and 5 °C; and 1 sun and 60 °C.

**Figure 13 sensors-19-01578-f013:**
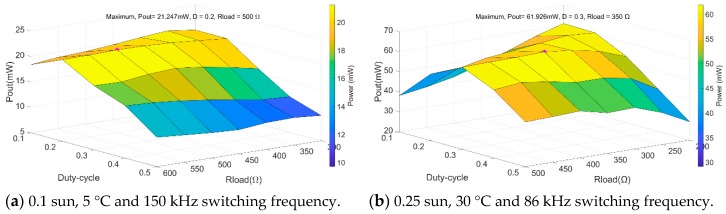
DC/DC output values dependent on different input and configuration conditions.

**Figure 14 sensors-19-01578-f014:**
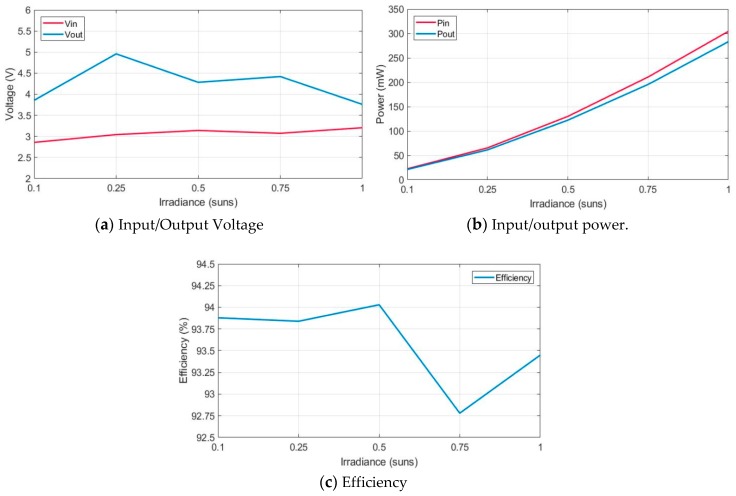
Converter sweep maximum results with different irradiance and temperature.

**Table 1 sensors-19-01578-t001:** SLMD481H08 solar module electrical characteristics [24].

Symbol	Cell Parameter	Value
V_oc_	Open circuit voltage	5.04 V
I_sc_	Shor circuit current	200 mA
V_mpp_	Voltage at max. power point	4 V
I_mpp_	Current at max. power point	178 mA
P_mpp_	Maximum peak power	712 mW
FF	Fill factor	>70%
η	Solar cell efficiency	22%
ΔV_oc_/ΔT	Open circuit voltage temp. coefficient	−2.1 mV/K
ΔJ_sc_/ΔT	Short circuit current temp. coefficient	0.12 mA/(cm^2^K)

**Table 2 sensors-19-01578-t002:** Maximum voltage, current and power obtained with simulation under different operation conditions.

G (sun)	T (°C)	R_load_ (Ω)	V_out_ (V)	I_out_ (mA)	P_out_ (mW)
0.1	5	380	2.913	7.666	22.333
0.1	25	360	2.937	8.157	23.956
0.25	10	160	3.167	19.792	62.675
0.25	30	155	3.224	20.803	67.077
0.5	15	80	3.204	40.044	128.281
0.5	45	75	3.258	43.437	141.511
0.75	20	55	3.324	60.439	200.908
0.75	40	50	3.275	65.502	214.525
1	22	40	3.304	82.606	272.948
1	60	35	3.277	93.627	306.812

**Table 3 sensors-19-01578-t003:** Comparison between simulation and test results values at different ambient conditions.

**Ambient Conditions**		**Mean Standard Deviation**	
**G (sun)**	**Temp (°C)**	**I (mA)**	**P (mW)**
0.25	46	1.760	5.056
0.5	65	6.625	18.254
		**Mean Percent Deviation**	
		**I (%)**	**P (%)**
0.25	46	3.148	3.009
0.5	65	2.316	2.395

**Table 4 sensors-19-01578-t004:** Comparative results between different MOSFETs at 0.1 sun and 5 °C.

MOSFET	Pcontrol (µW)	Rds (mΩ)	Qg (nC)	Vin (Vp)	Vout (V)	Iout (mA)	Pout (mW)	Pin (mW)	η (%)
Si1555DL_N [32]	61.252	0.63	0.8	2.221	2.674	6.686	17.884	19.438	91.94
Si8416DB [33]	1197.9	21	17	2.185	2.655	6.639	17.618	19.183	90.64
SiA436DJ [34]	1189.1	8.7	15	2.182	2.653	6.633	17.603	19.159	90.69
Si1553CDLN [35]	26.717	0.55	0.578	2.256	2.69	6.726	18.096	19.679	91.93
Si8466EDB [36]	592.41	6.8	0.05	2.246	2.685	6.714	18.032	19.608	91.37
Si3900DV [37]	156.68	130	2.1	2.261	2.694	6.735	18.144	19.711	91.89

**Table 5 sensors-19-01578-t005:** Comparative results between different MOSFETs at 1 sun and 60 °C.

MOSFET	Pcontrol (µW)	Rds (mΩ)	Qg (nC)	Vin (Vp)	Vout (V)	Iout (mA)	Pout (mW)	Pin (mW)	η (%)
Si1555DL_N	28.181	0.63	0.8	3.424	3.772	75.45	284.64	305.836	93.04
Si8416DB	487.35	21	17	3.416	3.778	75.568	285.532	305.964	92.83
SiA436DJ	476.33	8.7	15	3.415	3.778	75.57	285.548	305.984	92.84
Si1553CDLN	11.603	0.55	0.578	3.432	3.768	75.364	283.989	305.708	92.88
Si8466EDB	242.1	6.8	0.05	3.418	3.778	75.569	285.537	305.935	93.09
Si3900DV	64.318	130	2.1	3.421	3.777	75.548	285.382	305.884	93.23

**Table 6 sensors-19-01578-t006:** Comparative results between different diodes at 0.1 sun and 5 °C.

Diode	Vin (Vp)	Vout (V)	Iout (mA)	Pout (mW)	Pin (mW)	η (%)
MBR0520L [38]	2.221	2.674	6.686	17.884	19.438	92.01
BAT15-03W [39]	2.332	2.636	6.591	17.378	20.177	86.13
1PS79SB31 [40]	2.253	2.675	6.687	17.89	19.692	90.85
NSR20F30NXT5G [41]	2.268	2.688	6.72	18.066	19.759	91.43
NSR0320MW2T1G [42]	2.238	2.676	6.69	17.907	19.55	91.60
NSVR351SDSA3 [43]	2.273	2.669	6.673	17.815	19.79	90.02
STPS20L15G [44]	2.089	2.614	6.536	17.092	18.485	92.46

**Table 7 sensors-19-01578-t007:** Comparative results between different diodes at 1 sun and 60 °C.

Diode	Vin (Vp)	Vout (V)	Iout (mA)	Pout (mW)	Pin (mW)	η (%)
MBR0520L	3.221	3.761	75.229	282.974	304.822	92.83
BAT15-03W	3.456	3.502	70.046	245.324	305.18	80.39
1PS79SB31	3.254	3.731	74.635	278.525	305.531	91.16
NSR20F30NXT5G	3.226	3.756	75.129	282.222	304.939	92.55
NSR0320MW2T1G	3.236	3.748	74.97	281.028	305.166	92.09
NSVR351SDSA3	3.31	3.675	73.502	270.134	306.289	88.20
STPS20L15G	3.134	3.805	76.108	289.623	302.117	95.86

**Table 8 sensors-19-01578-t008:** Comparative results between different coils and frequencies at 0.1 sun and 5 °C.

L (µH)	T (µs)	fsw (kHz)	Vin (Vp)	Vout (V)	Iout (mA)	Pout (mW)	Pin (mW)	Pcontrol (µW)	η_total (%)
100	10	100	2.377	5.872	2.936	17.249	20.458	137.85	84.18
	20	50	1.611	5.168	2.584	13.354	14.614	62.802	91.32
	30	33.3	1.094	4.282	2.141	9.171	10.06	38.516	91.12
	40	25	0.938	4.03	2.015	8.124	8.659	29.024	93.79
	50	20	0.768	3.6	1.8	6.482	7.113	24.891	91.10
200	10	100	3.278	6.266	3.133	19.635	20.743	133.97	94.52
	20	50	2.386	6.192	3.096	19.175	20.512	60.406	93.42
	30	33.3	1.86	5.628	2.814	15.841	16.68	41.029	94.93
	40	25	1.604	5.19	2.595	13.471	14.549	30.029	92.56
	50	20	1.298	4.667	2.333	10.891	11.882	24.737	91.63
300	10	100	3.334	6.221	3.11	19.351	20.168	124.32	95.82
	20	50	3.076	6.518	3.259	21.245	22.034	66.122	96.35
	30	33.3	2.604	6.448	3.224	20.794	21.639	44.911	96.05
	40	25	2.12	6.021	3.011	18.127	18.712	34.287	96.84
	50	20	1.894	5.628	2.814	15.841	16.96	26.132	93.38

**Table 9 sensors-19-01578-t009:** Comparative results between different coils and frequencies at 1 sun and 60 °C.

L (µH)	T (µs)	fsw (kHz)	Vin (Vp)	Vout (V)	Iout (mA)	Pout (mW)	Pin (mW)	Pcontrol (µW)	η_total (%)
100	10	100	4.1	6.408	32.04	205.319	217.159	144.07	94.40
	20	50	3.785	7.228	36.143	261.27	276.827	76.571	94.30
	30	33.3	3.48	7.546	37.724	284.631	304.52	52.631	93.42
	40	25	3.263	7.512	37.561	282.178	305.686	33.705	92.28
	50	20	2.765	7.2	35.977	258.887	280.732	28.683	92.19
200	10	100	4.207	5.848	29.242	171.023	178.726	143.69	95.55
	20	50	4.127	6.327	31.639	200.212	209.853	73.559	95.33
	30	33.3	3.988	6.805	34.018	231.554	243.55	48.988	95.03
	40	25	3.89	7.208	36.04	259.786	277.011	36.973	93.74
	50	20	3.74	7.303	36.487	266.706	286.158	29.011	93.17
300	10	100	4.207	5.848	29.24	171.023	179.139	143.8	95.33
	20	50	4.218	5.813	29.067	168.984	176.696	71.312	95.56
	30	33.3	4.136	6.322	31.617	199.987	212.42	48.61	94.10
	40	25	4.009	6.678	33.391	223.001	238.667	37.604	93.40
	50	20	3.992	6.904	34.52	238.333	256.205	28.677	93.00

**Table 10 sensors-19-01578-t010:** Converter sweep maximum results with different irradiance and temperature.

Ambient Conditions		DC/DC Operation Conditions			Output Values				
G (sun)	T (°C)	f_sw_ (kHz)	R_load_ (Ω)	D	V_in_ (V)	V_out_ (V)	P_in_ (mW)	P_out_ (mW)	η (%)
0.1	5	150	500	0.2	2.858	3.856	22.633	21.247	93.88
0.25	30	86	400	0.3	3.043	4.958	65.51	61.476	93.84
0.5	15	66	150	0.3	3.141	4.283	130.061	122.292	94.03
0.75	40	45	100	0.3	3.074	4.42	210.78	195.57	92.78
1	60	25	50	0.2	3.206	3.76	304.45	283.18	93.45

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
