# Peer review of "Photovoltaic Energy Harvesting System Adapted for Different Environmental Operation Conditions: Analysis, Modeling, Simulation and Selection of Devices"

_sensors, 2019, doi:10.3390/s19071578_

Round 1

Reviewer 1 Report

This paper presents only basic simulation results, using datasheet info for an elemental solar cell modelling. The paper is merely an undergraduate exercice, by selecting the components for a simple photovoltaic system that is just simulated using basic tools.

Methodology presented in page 5 is elemental, and does not provide any information.

The work must be supported by consistent experimental results. Finally, references are extremely poor, most of them being just only datasheets.

Author Response

This paper presents only basic simulation results, using datasheet info for an elemental solar cell modelling. The paper is merely an undergraduate exercise, by selecting the components for a simple photovoltaic system that is just simulated using basic tools.

Apart of simulations, the photovoltaic cell harvester’s model has been justified with real measures which shown that it is appropriate.

Methodology presented in page 5 is elemental, and does not provide any information.

The commented methodology which was on page 5 has been eliminated, certainly it was a bit elemental.

The work must be supported by consistent experimental results. Finally, references are extremely poor, most of them being just only datasheets.

Experimental results have been included. Text, justifications, and explanations have been improved, and more consistent references have been included, too. Datasheets of devices have been maintained as references, but they could be omitted if it is seen necessary. In addition, English has been improved and an English native has revised the paper.

Reviewer 2 Report

The authors present an interesting study on photovoltaic energy harvesting system adapted for different environmental operation conditions.

The introduction should be much more developed and better referenced.

The axis labels and numbers of Figure 10 should be greatly enhanced. As it stands, it is really small.

English should be enhanced, for instance the sentence "The results corroborate that converter works properly in boost mode for all ambient conditions." can be stated in a much better way.

Table 9 and Figure 11 should be analysed in more detail.

Author Response

The authors present an interesting study on photovoltaic energy harvesting system adapted for different environmental operation conditions.

The introduction should be much more developed and better referenced.

The introduction has been modified and more referenced as suggested.

The axis labels and numbers of Figure 10 should be greatly enhanced. As it stands, it is really small.

The axis labels and numbers of Figure 10 have been enhanced as said and now can be seen.

English should be enhanced, for instance the sentence "The results corroborate that converter works properly in boost mode for all ambient conditions." can be stated in a much better way.

English has been improved and an English native has revised the paper.

Table 9 and Figure 11 should be analysed in more detail.

Table 9 and Figure 11 have been analysed in more detail as requested and the paragraph after them has been modified adding more descriptive information.

Round 2

Reviewer 1 Report

Thank you for including the proposed suggestions.

Currently the paper presents a more finished aspect. However, some considerations should be included:

According to the work, authors select a DC-DC boost architecture as PV module converter, selecting the suitable components to achieve a better performance. An important issue in boost-based converters is the look for the point in the PV power curve where maximum power transference can be achieved. This is done by controlling the switching frequency in the MOS transistor, so that the equivalent input impedance of the DC-DC varies to maximize the power transferred from the PV panel. In this work, this issue is slightly mentioned just indicating numerical values of frequency switching and a minimum comment in lines 238-239. Due the importance of a suitable selection of this parameter in the efficiency of the energy transference, it should be justified the selection of frequency beyond the mere numerical indication. Most works related to DC-DC conversion (specifically in PV applications), use the corresponding paragraphs to describe suitable algorithms or electronic circuitry applied to control the MOS transistor duty cycle (see, e.g. doi:10.3390/s17081794). Authors should deepen this aspect of their work.

Another issue addressed in this work is the variation of the Rload to improve the operation of the DC-DC converter. Authors should explain how it is possible to modify this load in a realistic operation, where the energy storage device can be a determined battery or supercap, and the final impedance value is given by the electronic device to be powered, whose equivalent impedance is not clear that can be modified at the user discretion.

Finally, some few typos:

Line 59 “for ITS storage”

Line 123: “Finally, the test STARTS and…”

Line 150: “3.009% in POWER”

Line 173: “Boost control techinques ARE based…”

Author Response

Thank you for including the proposed suggestions.

Thank you for your new comments, they have helps us to improve the article.

Currently the paper presents a more finished aspect. However, some considerations should be included:

According to the work, authors select a DC-DC boost architecture as PV module converter, selecting the suitable components to achieve a better performance. An important issue in boost-based converters is the look for the point in the PV power curve where maximum power transference can be achieved. This is done by controlling the switching frequency in the MOS transistor, so that the equivalent input impedance of the DC-DC varies to maximize the power transferred from the PV panel. In this work, this issue is slightly mentioned just indicating numerical values of frequency switching and a minimum comment in lines 238-239. Due the importance of a suitable selection of this parameter in the efficiency of the energy transference, it should be justified the selection of frequency beyond the mere numerical indication. Most works related to DC-DC conversion (specifically in PV applications), use the corresponding paragraphs to describe suitable algorithms or electronic circuitry applied to control the MOS transistor duty cycle (see, e.g. doi:10.3390/s17081794). Authors should deepen this aspect of their work.

Subsection named Control definition (3.1.1) has been included to describe this process and a few modifications in 3.4. In addition, several references, including the one suggested, have been added to support the work principles.

Another issue addressed in this work is the variation of the Rload to improve the operation of the DC-DC converter. Authors should explain how it is possible to modify this load in a realistic operation, where the energy storage device can be a determined battery or supercap, and the final impedance value is given by the electronic device to be powered, whose equivalent impedance is not clear that can be modified at the user discretion.

A paragraph that describes this process has been included at the start of section 3.5 (lines 266-274 in the pdf document).

Finally, some few typos:
Line 59 “for ITS storage”
Line 123: “Finally, the test STARTS and…”
Line 150: “3.009% in POWER”
Line 173: “Boost control techinques ARE based…”

Theses grammar errors have been corrected.

Reviewer 2 Report

A good review. Congratulations.

Author Response

A good review. Congratulations.

Thank you very much.

Round 3

Reviewer 1 Report

Thank you for attending the suggestions.